# From the Brain to the Field: The Applications of Social Neuroscience to Economics, Health and Law

**DOI:** 10.3390/brainsci7080094

**Published:** 2017-07-28

**Authors:** Gayannée Kedia, Lasana Harris, Gert-Jan Lelieveld, Lotte van Dillen

**Affiliations:** 1Department of Psychology, University of Graz, A-8010 Graz, Austria; 2Department of Experimental Psychology, University College London, London WC1H 0AP, UK; lasana.harris@ucl.ac.uk; 3Unit Social and Organizational Psychology, Institute of Psychology, Leiden University, 2311 EZ Leiden, The Netherlands; lelieveldgj@fsw.leidenuniv.nl (G.-J.L.); dillenlfvan@fsw.leidenuniv.nl (L.v.D.)

**Keywords:** social neuroscience, applications, health, law, neuroeconomics, reverse inference, reward, social exclusion, morality

## Abstract

Social neuroscience aims to understand the biological systems that underlie people’s thoughts, feelings and actions in light of the social context in which they operate. Over the past few decades, social neuroscience has captured the interest of scholars, practitioners, and experts in other disciplines, as well as the general public who more and more draw upon the insights and methods of social neuroscience to explain, predict and change behavior. With the popularity of the field growing, it has become increasingly important to consider the validity of social neuroscience findings as well as what questions it can and cannot address. In the present review article, we examine the contribution of social neuroscience to economics, health, and law, three domains with clear societal relevance. We address the concerns that the extrapolation of neuroscientific results to applied social issues raises within each of these domains, and we suggest guidelines and good practices to circumvent these concerns.

## 1. Introduction

Social neuroscience is an interdisciplinary field that aims to investigate how biological systems implement social behavior, and to understand social processes by examining their biological underpinnings [1]. Social neuroscience holds the promise of understanding people’s thoughts, emotions and intentions through the mere observation of their biology. If scientists were indeed able to establish an accurate correspondence between biological functions on the one hand and social cognitions and behaviors on the other hand, neuroscientific methods could have tremendous applications for other disciplines and society in general. Such realms of human behavior concerned with social interactions include economic decision-making and exchanges, physical and mental health care and prevention, and jurisprudence. In fact, scholars, practitioners and experts in these disciplines as well as the general audience interested in these topics are already relying on the insights and methods of social neuroscience to explain, predict and change behavior [2,3,4,5,6,7]. The increasing number of academic articles and research published in these applied domains are all the more influential when paired with neuroscientific evidence, especially when accompanied by brain images, and are perceived as more credible than other kinds of scientific evidence [8]. In the present review article, we intend to provide examples of the contribution of social neuroscience to the domains of economics, health and law, address the concerns raised by the extrapolation of neuroscientific results to applied disciplines, and suggest guidelines and good practices to circumvent these concerns.

The question of reverse inference is one of these concerns [9]. Reverse inference refers to the assumption that the engagement of a certain mental process (e.g., a preference for option A in comparison to option B) can be inferred from the observed activity of a biological system, such as the BOLD signal in a certain brain region. Brain activity has been used as an indicator of participants’ preferences for one presidential candidate over another [10,11] or for one soda brand over a similar one [12]. It has been used to make assumptions about the knowledge that someone holds and the sincerity of his or her claims [13]. Similarly, researchers have relied on reverse inference to predict the likelihood of a person developing unhealthy habits and to identify efficient methods to change them [14]. Reverse inference is a strategy commonly used to interpret results of social neuroscience studies.

The problem is that brain science still lacks understanding of cerebral functions to infer mental processes from neural activations. Such interpretations, therefore, often fall victim to a logical fallacy. That cognition X (e.g., the evaluation of a monetary reward) engages brain region Y (e.g., activation in the ventral striatum) is not sufficient evidence to assume that if a participant displays activation in region Y (e.g., when perceiving a presidential candidate), he or she is necessarily engaging in cognition X (e.g., a positive evaluation). Brain region Y could, for example, be associated with several different functions or it could be non-specific to the task investigated in the first place (e.g., it could reflect increased attentional mechanisms instead of reward processing). The use of reverse inference in social neuroscience literature thus arouses legitimate concerns.

If the question of reverse inference is of concern for the general field of cognitive neuroscience, it becomes even more acute in applied social domains as they usually involve high-level cognitive processes. One of the prerequisites for reverse inference is that the observed activations specifically reflect the mental process at stake. Now, theoretical constructs such as feelings of social rejection (known to have detrimental effects on health), fairness (of interest for behavioral economists), or deception (which elicits much discussion in the legal domain) are highly complex mental phenomena that encompass many basic cognitive and motivational processes. The tasks created to study these constructs may thus differ in many respects from the control conditions to which they are contrasted. They may require more attentional resources or the attribution of mental states to others; they may lead to the retrieval of autobiographical memories and elicit diverse emotional reactions or moral conflicts in participants. Thus, it is likely that these constructs rely on a wide range of unspecific brain activations. Moreover, numerous studies have demonstrated that brain responses to social stimuli are affected by the format in which these stimuli are presented (e.g., pictures vs. videos of faces; [15]), as well as by the level of expertise of the participants [16], their expectations [17], their personality [18] or the social environment in which they perceive these stimuli [19]. Thus, in addition to the functional specificity of the brain region under scrutiny, the predictive power of a reverse inference depends on the task setting and the population under investigation. This complexity renders the use of reverse inference particularly challenging in social neuroscience studies, as well as any attempt to extrapolate the results of neuroscience studies to real-life situations.

On that note, the ecological validity of neuroscience experiments is also an important issue to address. To state the obvious, most neuroscience methods impose heavy constraints on participants that prevent them from reacting naturally as may occur in real-world settings. Participants’ movements are usually restrained, the environment in which they perform the task may be loud and confined, and they are aware of being observed, which may generate unusual anxiety as well as impression management concerns and create experimental demand effects. Moreover, most neuroscience approaches require numerous acquisitions in order to maximize the signal-to-noise ratio. In a single session, participants are usually exposed to hundreds of trials consisting of the same task. This multiplicity of trials is often tedious and exhausting for participants and results in biases and cognitive processes that differ from the real-world situations for which they are supposed to account. Finally, in the vast majority of neuroscience studies, the social context is limited to the presentation of pictures, videos or vignettes describing other people and/or their actions, without any direct interpersonal contact or personal motivation. Studying social cognition in such an artificial context is, to say the least, a challenge for social neuroscientists who wish to relate their findings to real-world interactions.

Neuroscientists willing to consider the applications of their findings should also keep in mind the question of the representativeness of their sample [20]. Ninety-six percent of participants in psychology studies are from Western, Educated, Industrialized, Rich and Democratic (WEIRD) populations. Researchers often assume that their findings are universal, but comparative studies draw a much more complex picture; they report differences between industrialized and small-scale societies, Western and non-Western cultures, Americans and other Westerners, sub-populations of the American society, and contemporary Americans as compared to previous generations. Different populations display major variations in processes including visual perception, spatial reasoning, categorization, inferential induction, fairness, cooperation, and moral reasoning [21]. These cultural variations have been initially observed in behavioral studies. More recently, neuroimaging experiments have highlighted the neural networks underlying these differences [22]. Thus, social neuroscience research is mainly performed with WEIRD participants, known to be statistical outliers, although these findings are applied to populations mainly constituted of non-WEIRD persons, even in the West. Therefore, guidelines for good practices in social neuroscience studies should address the issue of the generalizability of their results.

The question of the generalizability and reproducibility of research findings is one that provokes important discussions among social neuroscientists. Like other areas of empirical science, research in social neuroscience could fall victim to questionable practices that undermine the validity of its results. Reliance on flexible statistical analyses, post-hoc alterations to hypotheses, low statistical power in neuroimaging studies, and the so-called file drawer problem—the tendency to publish positive results over negative or nonconfirmatory results—are some of the biases that could contribute to a distorted depiction of brain functioning. These issues are not specific to the field of social neuroscience [23,24]; all other areas of empirical science have to deal with similar challenges and face the same replication crisis. Nevertheless, they need to be addressed before making any claim regarding practical applications.

Our review focuses on the domains of economics, health and law, three applied disciplines that have, in recent years, extensively drawn from social neuroscience. Our intention here is not to provide an exhaustive overview of each of these domains—we would need several articles to cover such a vast array of subjects. Rather, our goal is to (1) illustrate how scholars use social neuroscience findings to address societal issues, (2) tackle the shortcomings and limitations of these approaches and (3) suggest guidelines for good practice. The research topics that we discuss in this article as examples deal with questions of central importance for our societies: how economic agents make decisions when they are embedded in a social context (see Section 2), the mechanisms by which people with lower social status are more at risk of suffering poor physical and mental health (see Section 3), and the cognitive biases that may affect legal professionals (see Section 4). Social neuroscience offers a wide palette of methods to address these questions, ranging from brain imaging to hormonal, cellular and genetic approaches. However, in this article, for reasons of concision, we will mainly focus on neuroimaging and electrophysiological studies, as these methods constitute the largest proportion of the research published in these domains. Although the field of economics, health and law investigate different theoretical questions, they have in common the use of neuroscientific research results for applied purposes and draw from social neuroscience studies that rely on the same methods. This means they often encounter the same challenges, such as those mentioned above (i.e., reverse inference, ecological validity, sample representativeness, and replicability). Our review intends to illustrate and address these domain-general issues as well as some of the pitfalls more specifically related to each of the applied fields that we examine.

## 2. Economics

In the late 1990s, economists began to pay attention to neuroscientific accounts of emotional and social decision-making (see e.g., [25]). This led to the rise of a new field—neuroeconomics—that attempts to integrate ideas and methods from psychology, economics and cognitive neuroscience with the goal of producing more robust models of economic decision-making (see [26,27]). This field has been dominated by a branch of experimental economics known as game theory. Game theory offers a rich variety of monetary games to study a wide range of economic and psychological phenomena, such as social and economic exchange, trust, prosocial behavior, self-interested tendencies and coordination [28].

To give an example, one of the most widely used games in economics literature is the Ultimatum Game (UG, developed by [29]). The UG has often been used to study strategic behavior and fairness. Two players decide on the distribution of a sum of money. The proposer (one of the two players) can specify the distribution and the responder (the other player) can either accept or reject the offer. If the offer is accepted, the money is divided accordingly, if the offer is rejected none of the players receives anything. Several studies in the field of social neuroscience and neuroeconomics have used the UG (e.g., [30,31,32,33,34,35]). Indeed, the UG offers the advantage of possessing a simple structure that makes this game compatible with the high number of trials required for neuroimaging, and at the same time enables a varied and rich pattern of decision-making. As we will see later in this section, neuroeconomists have used these advantages and extensively relied on the UG to investigate the neural correlates of decision-making in a social context.

Social neuroscience has offered a new way to measure the basic psychological processes that underlie economic decision-making without inferring them from behavior, and this research has provided new insights. First, neuroscientific studies have shown that some economic choices assumed to be different in theory in fact rely on the same brain circuitry and thus may involve similar cognitive mechanisms. Second, other research has found that seemingly similar economic behaviors engage different brain mechanisms and may thus may differ more than economists initially assumed. Finally, by showing changes in behavior when brain regions do not function optimally, we can draw conclusions about the mechanisms involved in functional economic decision-making. We will use examples from the fields of reward processing, fairness and cooperation to discuss these contributions.

Economists have long studied phenomena related to reward processing, such as reinforcement learning (learning through penalties and incentives [36]) and decision-making [37]. In the last decade, economists started incorporating insights from social psychology to investigate how the social context influences reward processing (e.g., [38]). Neuroimaging studies have shown that the brain reward system reacts to social rewards (e.g., positive emotional expressions by others, or social approval in the form of a thumbs up) in similar ways as to monetary rewards; both social and nonsocial rewards showed increased activation in the ventral striatum, ventromedial prefrontal cortex (vMPFC) and orbitofrontal cortex (OFC) [36,39,40,41]. In the economic trust game (an economic investment game designed to measure trust, see [42]) being perceived as trustworthy activates brain regions that are involved in reward processing, such as the striatum [43,44]. Similarly, gaining social approval by donating money [45] and viewing smiling faces in an economic game [41] have been shown to engage these same regions. Thus, while different at first sight, social rewards (e.g., being labeled trustworthy, gaining social approval and viewing smiling faces) and nonsocial rewards (e.g., money) seem to rely on similar brain mechanisms.

While some of the underlying neural mechanisms for processing social and non-social rewards seem to be similar, there are also important differences. Reward and feedback coming from other people engage social cognition brain systems and striatal regions to a greater extent than reward and feedback from a computer [46], supporting the conclusion that the social context affects reward and economic decision-making. In one study, researchers compared the processing of shared monetary rewards with a close friend to monetary rewards shared with a stranger or a nonsocial entity (a computer). The results revealed that monetary gains shared with a close friend enhanced subjective ratings and ventral striatal activation compared to the two other types of reward [47]. In line with this work, one of the most telling neuroscientific findings using economic games showed that receiving unfair offers from others in the ultimatum game [29] leads to increased activation in the prefrontal cortex, the anterior cingulate cortex and the anterior insula [33]. More important for our reasoning, they also showed that unfair offers made by human agents activate the bilateral anterior insula to a greater extent than the same offers made by computer agents. These differences in brain activity for human and computer agents highlight that social decision-making relies on different neural processing than non-social decision-making.

Finally, we can turn to neuroscientific measures to find out why people make different (often suboptimal) economic decisions, by comparing individuals with optimal brain function to individuals who have lesions in the brain or whose brain are not yet fully developed. Researchers [48] studied rejection rates in the ultimatum game among ventral medial prefrontal cortex (VMPC) lesion patients and control subjects, and showed that VMPC lesion patients showed an increase in rejection rates, which correlated with the perceived unfairness of the offers. Moreover, neuroscientists [49] examined developmental changes in brain regions associated with reactions to unfair offers in the ultimatum game. In adults, reactions to unfairness were not only affected by the offer itself but also by the ascribed intentionality of the person making the offer; unintentional unfair offers lead to increased acceptance rates. Younger adolescents, however, were less affected by intentions; they showed a higher rejection rate of unintentional unfair offers, which was associated with decreased activation in the temporoparietal junction and dorsolateral prefrontal cortex, two regions known to mature later in adolescence.

Although the use of social neuroscience has greatly advanced knowledge on economic decision-making, the field also still faces some limitations. The field of economics has notably long operated with a ban on using deception [50]. For example, in other areas of psychology experimenters inform participants that they are playing games with another participant when in fact, the other player is a computer or a confederate, or participants receive a different amount of money than announced at the beginning of the experiment. If participants know that deception is a common practice in experimental studies, they may stop believing the cover stories experimenters tell them. In the field of social neuroscience, deception is mostly considered permissible but social neuroscientists who study economic decision-making may see their manuscripts rejected when reviewed by economists or submitted to economics journals. These opposing opinions about whether deception should be permissible may be the reason why combining insights from social and economic neuroscience is difficult. One way to resolve this issue is for social neuroscientists to generate new experimental paradigms where they avoid using deception. They could do so, not only because deception could influence the results directly, but also because this could lead to a better integration of the fields of economics and social neuroscience in economic decision-making. Previous work that incorporated the social context, for instance, invited real partners to the scanner room where participants played economic games [51], instead of wrongly informing participants about the presence of these partners. Although avoiding deception can be challenging for social neuroscientists, such a strategy may facilitate the integration of social neuroscience and economics findings.

The question of reverse inference is also an issue that concerns neuroeconomics, but the literature offers ways of dealing with this problem. One approach that has often been used in the neuroeconomics literature—as well as in the other applied areas discussed in this article—is to implement a theory-driven experimental design by directly manipulating the underlying process. Here, the aim is to contrast conditions that only differ in the cognitive process of interest for the researchers and/or to measure this cognitive process with self-reports or behaviors. Although this approach does not eliminate the logical fallacy behind reverse inference, it enables neuroscientists to interpret brain activations with more confidence. Another solution is to conduct meta-analyses. Meta-analyses using Bayesian statistics enable neuroscientists to estimate how specific the activation of a particular region of the brain is for a given cognitive process [52,53]. An example of such an approach can be found in a recent publication [54]. The authors of this article estimated the probability for a reward process given the observation of nucleus accumbens (NAc) activation. They found that NAc activation increases the probability of a reward-related process taking place to 0.90 (odds 9:1). This high probability demonstrates a moderate to strong relationship between reward processes and NAc activity and, thus, legitimates the use of reverse inferences in this context.

Finally, in the field of social neuroscience, more so than in the field of economics, most conclusions are based on brain data from university students, which may poorly reflect the learning and experience of “real-world” economic decision-makers. Indeed, one study that compared neural processes involved in economic decision-making between managers and non-managers found differences in subcortical activation (e.g., caudate activation), suggesting that the underlying processes for making economic decisions may be different for more experienced decision-makers [55]. Future studies could include managers, traders, and/or bankers as potential participants in neuroeconomic studies on decision-making to increase ecological validity and sample representativity.

## 3. Health

Over the past few decades, it has become clear that the social context has a pervasive influence on health outcomes. Our (perceived) social standing, the quality of our social interactions, and the empathy and support we receive from others all form possible sources of stress, and accordingly determine our psychological and physical wellbeing [56]. In the last two decades, researchers have begun to incorporate neuroscientific measures in their study of these influences. This evolution was, at least in part, inspired by the insight that self-reports and behavioral measurements of rejection, stigma and loneliness are often subject to strong social norms and may not always be accessible to introspection [57]. Neuroscientific measures are certainly not the only means to circumvent the issues that explicit measures raise. There exists, for example, a rapidly growing literature on the cardiovascular consequences and endocrinological effects of psychosocial factors on health outcomes [58,59]. Neuroscientific methods have certain disadvantages and advantages over these methods. Cardiovascular and endocrinological responses are directly tied to the health outcomes under examination, such as elevated blood pressure and chronic stress, whereas neuroscientific indices typically only provide indirect evidence of such psychophysical states. However, neuroscience methods allow for the examination of relatively fast stimulus-based responses, whereas cardiovascular and endocrinological measures are much better suited to examine more gradual changes in people’s psychological states. Social neuroscience may, thus, provide unique insights in how social stressors affect individual wellbeing.

Social stressors such as stigma, loneliness, and low social standing have a considerable impact on people’s psychological but also physical wellbeing (see [60] for an overview). People with relatively low socioeconomic status, for example, are more likely to develop physical and psychological illnesses and die prematurely [61], an effect that cannot be fully accounted for by differences in material resources, literacy or lower quality healthcare. Moreover, stigmatized groups experience higher stress levels, which wear out the cardiovascular system and lead to higher morbidity and mortality [62].

Social neuroscience research has demonstrated that repeated experiences with stigmatization or other forms of social threat sensitize the brain to detect and process cues that signal rejection [54], which in turn may make the brain more vulnerable to such social stressors. One functional magnetic resonance (fMRI) study found that loneliness is associated with greater visual cortex activation in response to social threat cues [63]. In another fMRI study [64], undergraduate students who retrospectively perceived their parents as having low social standing showed stronger amygdala activation to threatening faces compared to students who perceived their parents as having a higher social position. Interestingly, previous research has demonstrated that the limbic system, including the amygdala, modulates the activity of the hypothalamic-pituitary-adrenocortical (HPA) axis. The HPA axis coordinates stress responses through its control over hormonal outputs, and accordingly influences a wide range of physiological, behavioral, and health outcomes (e.g., [65]). Social standing may thus influence health outcomes through the activation of a path sequentially involving the limbic system, the HPA axis and the endocrine system. Moreover, the extensive neuroanatomical and functional connectivity between the prefrontal cortex and the limbic system allows for the modulation of stress reactivity by environmental appraisals, including assessments of the quality of relationships and social support [66]. This suggests that the perception of social threat, rather than actual threat, determines people’s sensitization to rejection, and their stress reactivity [58,67].

Social stressors may not only influence people’s psychological and physical wellbeing directly; they may do so through the behaviors they evoke, such as overeating, substance abuse and risky decision-making [68]. Social stressors may lead to poorer health-related decisions through their negative influence on self-control processes, either during, or following the experience of a stressful situation [57]. Perceived isolation has been associated with decreased impulse control [69] and with lower activity in the parietal and right prefrontal cortex [70]. Event-related potential (ERP) research [71], moreover, showed that women who had faced stereotype threat subsequently displayed reduced self-control, such as eating more ice cream, as well as impaired performance on a cognitive control task, such as a Stroop color-naming task. This latter effect was accompanied by amplified, non-distinct anterior cingulate cortex (ACC) activation throughout the task, suggesting that participants monitored their performance more stringently, regardless of whether cognitive control was actually required, which could have made them more vulnerable to self-control failure. Although none of these studies directly examined the long-term health consequences of self-control failure, current evidence suggests that the chronic presence of social stressors contribute to suboptimal health through their ongoing negative impact on self-control resources.

Social neuroscience, finally, may also inform how social stressors influence health outcomes through their effects on healthcare interactions. Health care providers may hold (non-conscious) biases towards members of certain groups that may lead them to treat some patients differently than others [72]. ERP research has demonstrated that social categorization occurs rapidly [73], and, at least in part, unintentionally, with differentiations based on race, for example, already appearing at around 120 ms after a face is presented (see [74] for an overview). People, moreover, display less neural resonance with the pain of an outgroup member [75] or a member of a stigmatized group (e.g., an obese compared to a normal-weight patient [76]), especially when this person is held responsible for his/her condition (i.e., when a patient contracted HIV through intravenous drug use [77]). Together, these social neuroscience findings suggest that social perceptions by health care professionals may shape their interactions with patients, and accordingly, the treatment patients receive.

So far, we have discussed several ways in which social neuroscience has contributed to a better understanding of the psychosocial factors influencing health outcomes, by revealing underlying processes that are not easily revealed through self-reports or behavioral assessments because of people’s lack of insight into these factors, or because of their fears of negative social evaluations. With most of these processes taking place outside of people’s awareness, the impression may arise that they are hard-wired, and not easily changed. This would, however, not only be dysfunctional, but also lead to faulty conclusions.

Through network and connectivity analyses, in the past decade, neuroscience research has uncovered mechanisms by which people are able to regulate their behavior so that the influence of automatic responses and biases on the decision-making process is reduced (see for example [78]). A research performed with newly arrived Chinese students in Australia provides a good illustration of factors that lessen biases towards outgroup members [79]. In this study, participants were presented with same-race (Chinese) and other-race (Australian) targets receiving painful stimulation. Although participants reported similar painfulness ratings for both kinds of targets, their brain data revealed a stronger reactivity to the same race targets in regions associated with pain perception (involving the insula and anterior cingulate cortex). Interestingly, this race bias was smaller the longer the Chinese students had lived in Australia. Moreover, contact frequency with Australians seemed to be a more important determinant of the bias than contact quality. Another study [80] demonstrated a similar race bias in pain perception using electroencephalography (EEG), again in the absence of any behavioral effects. This study, however, failed to find any beneficial effect of recategorization—participants were induced to categorize other race targets as ingroup members by means of a minimal group manipulation—another strategy widely studied in social psychology to reduce intergroup bias. Insights such as these thus may lead to a better identification of successful (versus less successful) coping mechanisms for individuals to deal with social stressors as well as concrete improvements in doctor–patient interactions. Thus, social neuroscience findings could not only be taken as evidence that certain social perceptions and biases lead to detrimental health outcomes, they may also aid the development and testing of effective interventions to reduce these influences.

However, one important limitation of social neuroscience research in the health domain involves its reliance on anatomical overlap in activation patterns as evidence for shared neural networks. For example, a great deal of literature has addressed the detrimental effects of social rejection on mental and physical wellbeing, by pointing out how it relates to a similar fMRI activity pattern to that of physical pain, most notably involving the dorsal anterior cingulate [81]. Accordingly, an important argument of this “shared representation” theory is that the anatomical overlap represents “distress” in a modality-independent way, applying to both social, and non-social situations. More recently, this idea has been challenged. Using pattern classification techniques, it was shown that whereas rejection and physical pain engaged overlapping brain regions, the rejection classifier performed at chance on nociceptive stimuli, and vice versa [82]. Thus, whereas areas involved in the pain matrix may not be specific to physical pain, certain patterns of activity in this network are. Ongoing research now addresses which interventions affect these patterns directly (e.g., distraction) versus indirectly (e.g., placebo, self-regulation), for example through the valuation of the nociceptive stimulus in a distinct network involving the nucleus accumbens and the ventromedial prefrontal cortex [83]. Approaches such as these provide important extensions to the interpretation of activity overlap, and are especially relevant in the health domain, where more precise neural activity patterns may help to accurately infer what factors affect individual suffering, even when the individual is unable or unwilling to report on this [84]. For the development and testing of such more fine-grained neural markers, access to large samples is crucial as models need to be developed and tested on separate datasets. Luckily, new means for large-scale scanning and data sharing are developing at a rapid pace [85].

As a final note on health, we wish to point out that neuroimaging findings are correlational in nature, which can have consequences for the conclusions drawn from (epidemiological) research on group differences, for example as the result of stigma or isolation. Unless lesion studies or transcranial magnetic stimulation can be implemented, this makes it difficult to draw causal inferences. In addition, there is always the possibility that differences in health outcomes between groups arise from differences in their actual physiology [86] rather than these being the result of social dynamics. Finally, as with epidemiological research more generally, possibly confounding variables such as income and education need to be carefully considered.

## 4. Law

Social neuroscience research has expanded our understanding of concepts relevant to law, including punishment and moral reasoning. However, the focus of social neuroscience actually used in the courtroom has been on lie detection. A number of companies claim to be able to use EEG and fMRI technologies to determine whether a person is lying or telling the truth [87,88]. However, lie detection is not a reliable application of social neuroscience to law because of the philosophical arguments surrounding lying; a person can be conveying false information that they believe to be true, thereby misrepresenting the truth [89]. People can also convince themselves that something that is false is indeed true [90]. Additionally, many people have figured out how to subvert physiological measurement during lie detection, suggesting that even if these philosophical issues can be dealt with, there is still the danger of people finding ways to subvert the technology. Finally, lying is a complex concept that may not be reducible to brain regions or networks. Therefore, any approach using brain imaging or measurement to assess lying may fall prey to the reverse inference problem because the context used to elicit the lie may not generalize. Even though social neuroscience can provide a glimpse into the black box of cognition, it is still not at the point where it can make a useful contribution to the law in this way.

Perhaps the real usefulness of social neuroscience for the law lies with illuminating the complex psychological constructs central to legal decision-making. For instance, much of legal decision-making is concerned with figuring out someone’s mental state [91]. The law punishes actors who have bad minds, or harbor ill intention and act upon it in subverting the law. Social neuroscience has identified the cognitive architecture underlying our ability to infer the mental states of others. This work has also highlighted how things such as biological information that draws focus away from the person’s mind (such as stating that they have a hormonal imbalance) can bias this architecture [92]. The science suggests that if the mind of the accused and the victim is not salient, legal decisions can change. This finding even holds for federal judges in the United States [93]. This is consistent with psychological research demonstrating the impact of emotion on legal decision-making. Evidence such as gruesome crime scene photographs, heartbreaking victim statements, and personal information about the defendants’ history may arouse strong emotions, which in turn influence judgments and decisions by a judge, jury or audience in favor of one party or the other (see [94]).

Social neuroscience has also elucidated the biological mechanisms behind punishment. Punishing the perpetrator of a crime has been shown to recruit brain regions beyond those implicated in higher order cognition, such as the amygdala, which activity correlates with punishment severity when participants read and judge vignettes of crimes [95,96]. One motive for punishment is communication; punishment signals to others that there are consequences if the law is broken. However, punishment systems in the brain extend beyond the amygdala, including the insula, areas of prefrontal cortex, and striatum [97]. These activations perhaps align with other motives for punishment, including restoration of the victim and retribution [98]. Therefore, the concept of punishment may not be homogeneous within the brain, and may actually rely on separable systems depending on the motives involved. The specific social context may lead to not only a more complex definition of punishment, but also better specificity related to its motives. Social neuroscience provides such insight, not simply in the case of punishment, but other legal concepts as well. For instance, one can ask whether different states of mind correspond to different brain structures or networks. This is not to say that such approaches will rely on reverse inferences from brain function, but can simply highlight whether there is overlap or difference between psychological processes (though even overlap should not lead to inferences of similar processes, as we discuss in our discussion of health).

Some social neuroscience suggests dissociated brain systems when people make different kinds of moral decisions. To investigate this question, brain researchers used the trolley dilemma from philosophy, a thought experiment in which one must choose between killing one person to save many lives, or letting many others die. The researchers suggest that making utilitarian decisions—deciding to sacrifice one life in order to save the others—engages brain systems associated with logical decision-making. On the contrary, deontological decisions—harming is wrong, so not even one life should be sacrificed, even to save more lives—engaged brain regions associated with emotional processing [99,100,101]. Though these interpretations rely on reverse inferences, they at least suggest that different types of moral decisions may rely on different brain regions. Additionally, traditionally dehumanized groups were sacrificed more and saved less, and decisions involving sacrificing them engaged the ACC, perhaps to override the conflict necessary to turn a default deontological decision into a utilitarian one [102]. However, other research suggests that damage to prefrontal regions increases utilitarian decisions [48]. Therefore, there is a lack of consensus about the mechanisms that govern moral reasoning.

The results just discussed fit with an affect-cognitive distinction popular in legal reasoning. Is people’s behavior driven by emotion (such as in crimes of passion), or do people intentionally plan and execute behavior (such as in premeditated murder)? This affective-cognitive distinction proposed during legal and moral reasoning is supported by several studies that utilize legal vignettes. Plain (non-emotion) versus colorful (emotional language) as well as intentional versus unintentional harms would modulate activity in affective and cognitive brain regions differently [103,104]. However, such a distinction runs up against the problem that cognition and affect are not clearly distinct constructs, but can rely on the same general underlying brain mechanisms [105]. Moral reasoning, for example, often integrates both types of information to solve complex problems [106,107]. Descriptions of crimes involving bodily harm modulate the social cognition network even when this information should be irrelevant to the decision, suggesting that gruesome details (emotional information) affect legal reasoning [108]. Despite this contribution, this study nonetheless does not account for the concern of ecological validity, since participants were deciding whether the sentences relating to the crimes made sense or not, instead of determining someone’s guilt or innocence. Therefore, though neuroscience holds the promise of resolving such legal and moral debates, the evidence typically complicates the debate further, motivating new social neuroscience research.

Social neuroscience holds the promise of illuminating many epistemologies because of its ability to cut across disciplinary boundaries, asking questions not apparent when simply considering one academic discipline. Why then has social neuroscience not played a more formal role in legal reasoning and judgment? Perhaps these studies often highlight flaws and biases in legal reasoning and punishment decisions, and do not clearly present evidence in favor of one side or another in an adversarial system. Such evidence would rely primarily on causal claims that could be drawn from brain analysis patterns or neural anatomy. However, the state of research in social neuroscience makes such causal claims difficult since complex cognitive processes such as those relevant to legal decisions may not be easily inferred from the activation of brain regions [9]. Moreover, the specific psychological functions and anatomical boundaries of brain regions are still often debatable, making any findings drawn from such activation patterns inconclusive.

There also remain cultural differences between legal scholars and practitioners, and social neuroscientists, presenting perhaps the greatest challenge when considering the benefit of social neuroscience for the law and legal studies. Most legal professionals hold strong biases against the utility of science, technology, engineering and mathematics (STEM) fields in the courtroom (beyond a select few), and may consider this domain of knowledge beyond their intellectual expertise. As such, they exhibit bias against such approaches when applied to the courtroom, as witnessed by the ruling in the United Kingdom to ban the mathematically based Bayesian network analysis (Bayes nets) from the courtroom because of the difficulty in understanding Bayes nets and the underlying statistical analyses [109]. In this instance, the defense reasoned that a Bayesian analysis of the strength of the evidence would suggest an outcome consistent with the client’s innocence. However, this strategy was deemed inadmissible because of the complicated nature of the underlying statistics.

Neuroscience is equally complicated, perhaps explaining why it also has not greatly influenced legal decision-making. But does this suggest then that social neuroscience has no use in the legal domain? Not entirely since the law readily accepts other types of scientific evidence into the courtroom, including DNA evidence, ballistics, and coroners’ reports. These aspects of science today often provide invaluable pieces of evidence, suggesting that there is a place for science in the courtroom. Furthermore, social neuroscience promises to inform how people make inferences and other complex decisions, the very abstract, unobservable constructs on which legal theorizing hinges. Social neuroscience may, in the future, be able to make such contributions if social neuroscientists continue studying phenomena directly relevant to legal decision-making. Therefore, though social neuroscience may not be at the stage where it can be admitted as evidence in the courtroom, it can influence legal strategizing and subsequent legal outcomes through a better understanding of how legal decision-making unfolds.

## 5. Conclusions

The present article describes how social neuroscience has contributed to three applied domains and emphasizes the importance of considering the social context for studying economic, health-related and legal decision making. Although not exhaustive, the research questions we reported as illustration were chosen because their theoretical background and the methodological approaches on which they rely are typical for these fields. The research reviewed in this article shows that different brain mechanisms are engaged when other people are involved and these mechanisms are modulated by the identity, the behavior, and the emotions elicited by the people with whom one interacts [41,42]. Therefore, considering the social context is of central importance for the ecological validity of the paradigms used in applied domains. 

In comparison to behavioral approaches, the neuroimaging and electrophysiological studies described here present significant advantages. They constitute implicit methods that enable to investigate unconscious processes as well as cognitions that participants may be unwilling or unable to explicitly admit. Social neuroscience research thus turns out to be of particular usefulness to explore participants’ motivations without having to rely on measurements that may be colored by self-serving biases and social desirability (e.g., when perceiving the member of a stigmatized outgroup or when punishing the culprit of a crime; see [77,97]). Moreover, these studies represent powerful tools to test overlaps and differences in brain activations and their putative underlying cognitive mechanisms, as illustrated by the similarities in brain activation in response to social and non-social rewards [41] or the differences reported between utilitarian and deontological moral decisions [101]. Finally, ERP approaches enable to look into social phenomena with a millisecond precision, a temporal resolution that few other methods can claim, revealing, for example, that social categorization takes place at surprisingly early stages of face processing [73]. The studies described in this article thus illustrate how social neuroscience results complement and sometimes even shed a new light on the cognitive and affective processes at stake in applied contexts. We showed how these findings can improve people’s financial decisions, promote better healthcare and fairer legal judgments. However, they also raise concerns that we intended to address.

The issue of reverse inference is transversal to all applied disciplines. Neuroeconomists, for example, draw inferences about the value that people attribute to a reward by observing the activity of the ventral striatum [29]. Similarly, in the health and legal domains, neuroscientists infer from the activity of the amygdala the perception of a social threat [60] or the desire to revenge and punish the culprit of a crime [95]. The validity of reverse inference is an ongoing debate but several approaches have been brought forward to improve confidence in reverse inferences. The first one consists in estimating the evidence of an association between a cognitive function and the activity of a brain region by running meta-analyses. Meta-analyses offer many advantages. They enable us to test the robustness of a result by providing statistical evidence across many studies and large samples and to assess the existence of a publication bias [110]. Researchers rely on meta-analyses to investigate the involvement of a certain brain region or network in a given task, the overlap or differences in brain activation between different tasks, or the role of moderating variables [111]. The more data there is, the higher the accuracy of estimates and the statistical power to detect an effect. Meta-analyses using Bayesian statistics can also provide information regarding the selectivity of the association between a cognitive function and activity in a certain brain region. They do so by taking into account not only the studies investigating the function in question but also those reporting activity in the same brain region for other tasks or assumed cognitive processes [48,53]. With regards to the applied domains reviewed in this article, Bayesian meta-analyses suggest a significant and selective association between reward representation and activity in the ventral striatum, as described in the section on neuroeconomics [50]. Researchers interested in other brain regions and cognitive processes should thoroughly check the strength of the evidence before making any reverse inference.

If meta-analyses constitute an improvement in comparison to qualitative reviews of the literature and an important step towards more reliable findings in social neuroscience, they also entail important pitfalls [53]. First, the results of the fMRI studies reported in such meta-analyses are correlational in nature and, therefore, do not provide any information regarding the causal role of a brain area in a cognitive process. Moreover, Bayesian statistics and the so-called Bayes factor are strongly influenced by prior parameters established by the researcher when he or she conceived the statistical model. Reverse inferences based on this approach, therefore, rely on strong assumptions that call for caution when claiming that a brain activity selectively represents a cognitive process.

An alternative approach to formally infer mental states from neuroimaging data relies on the use of multi-voxel pattern analysis (MVPA). This method is especially useful to test whether two tasks (e.g., tasks eliciting physical pain and social rejection as mentioned in the section on health) engage the same neural substrates and test the hypothesis of a shared mechanism [77]. One trains a classifier to discriminate between two conditions (e.g., nociceptive heat vs. neutral warmth stimulation) and tests whether it enables to categorize the supposedly related process (e.g., social rejection). Pattern classification techniques can, thus, provide valuable complementary information to meta-analysis results.

Although social neuroscientists will not be able to eliminate, in the near future, the constraints imposed by the technical component of their research, they can use a wide array of methods and approaches that complement each other and come with different advantages. Results from studies relying on EEG, functional MRI, brain stimulation or the investigation of patients suffering from a defined brain lesion, to name a few of these methods, need to be combined with behavioral approaches and compared with each other so that scientists can develop accurate models of brain functioning that can be applied to other disciplines and real-life situations. There is definitely more than one step from the brain to the field, and many social neuroscientists are willing to take the leap.

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
