# Peer review of "From the Brain to the Field: The Applications of Social Neuroscience to Economics, Health and Law"

_brainsci, 2017, doi:10.3390/brainsci7080094_

Round 1

Reviewer 1 Report

I found the promise of this manuscript to be highly engaging--a comprehensive review of social neuroscience contributions to date and in so many relevant domains is critically needed. Such a manuscript would be one of a kind. I did appreciate the section on Law where the issue of lie detection and it's poor application was discussed.

Unfortunately, I struggled to see how much of the rest of this manuscript delivered on its promises made upfront. For example, in many places it was difficult to understand whether the authors were presenting their own opinion or characterizing the opinion of others which they then wanted to refute. The ms talks about how brain activity can be used as a predictor of preferences for a presidential candidate, etc. But do the authors really believe that brain activity can be used to predict the preferences of an individual person? Or were they just trying to state that these claims have been made but need more careful scrutiny? I couldn't tell as a reader.

Another example of where more clarity was needed is the discussion of the focus on WEIRD samples. Are the authors arguing that there would actually be different brain processes in non-WEIRD samples for the very same psychological process in WEIRD samples? Or are they making a behavioral point which is that non-WEIRD samples may respond differently and therefore may engage brain systems that index the difference in response? This latter point doesn't seem like a point about how neuroscience contributes to our understanding? And the former point seems to go beyond the scope of what we know about biology? This same type of issue came up in the Economic Decision-Making section in which the ms describes how interacting with a friend is different than a computer. It was not clear how neuroscience was important here or told us anything different than the behavioral data. The ms references that there was enhanced ventral striatal activation but that is simply the same neural region typically associated with reward, so I wasn't sure how that contributed anything beyond what the enhanced subjective ratings told us). T

he Economic Decision-Making section (and also the Contribution of Social Neuroscience to Health) seemed focused on stating why social context and emotions needed to be taken into account but it wasn't clear how this was related to the thesis of showing how neuroscience contributes to psychology. Instead it seemed like a case for how psychology contributes to economics or health and no brain data was needed. A quick review of topic sentences in those sections illustrate that the writing doesn't even seem to be focused on anything about neuroscience's contribution. I found it surprising that the paper recommends that social neuroscientists should avoid using deception when published data has shown that deception doesn't make a difference in behavior.

The paper rightfully raises reverse inference as a problem but then falls victim to it. For example, on p. 4, the ms notes that caudate activation indicates that managers were using a heuristic decision strategy appropos of nothing about the actual manipulation but instead based on viewing caudate as a marker of automaticity. So I didn't understand how these examples help us learn about how neuroscience contributes to our knowledge of decision-making.

These types of interpretation and focus problems are riddled through the mansucript. My sense is that there is a lot of useful information here. A revision should focus on stating in what ways *social neuroscience*  contributes (or has tried to contribute and failed) to these domains rather than discussing how social and emotional contexts are important.

Author Response

Thank you for the opportunity to revise our manuscript. Below, we respond to the reviewers’ comments in detail. Such comments are italicized, while our responses remain in standard, non-italicized font.

Reviewer 1

I found the promise of this manuscript to be highly engaging--a comprehensive review of social neuroscience contributions to date and in so many relevant domains is critically needed. Such a manuscript would be one of a kind. I did appreciate the section on Law where the issue of lie detection and it's poor application was discussed.

Thank you for this positive feedback.

Unfortunately, I struggled to see how much of the rest of this manuscript delivered on its promises made upfront. For example, in many places it was difficult to understand whether the authors were presenting their own opinion or characterizing the opinion of others which they then wanted to refute.

We thank the reviewer for this observation. Indeed, we agree that we could have communicated our opinions more clearly. As such, we have re-written the entire manuscript. We now explicitly state our opinion in the introduction, as well as in each of the subsequent subsections.

The ms talks about how brain activity can be used as a predictor of preferences for a presidential candidate, etc. But do the authors really believe that brain activity can be used to predict the preferences of an individual person? Or were they just trying to state that these claims have been made but need more careful scrutiny? I couldn't tell as a reader.

We think that these claims have been made, but require more careful scrutiny. We state this explicitly on pages 4, 10, 21-22.

Another example of where more clarity was needed is the discussion of the focus on WEIRD samples. Are the authors arguing that there would actually be different brain processes in non-WEIRD samples for the very same psychological process in WEIRD samples? Or are they making a behavioral point which is that non-WEIRD samples may respond differently and therefore may engage brain systems that index the difference in response? This latter point doesn't seem like a point about how neuroscience contributes to our understanding? And the former point seems to go beyond the scope of what we know about biology?

We thank the reviewer for pointing out this lack of clarity. Our point concerning WEIRD populations was simply that until brain imaging samples are more representative of the global population, all findings could fail to generalize because of the dynamics of WEIRD populations. Specifically, we think that non-WEIRD samples may employ different behavior and brain mechanisms, but this is unknown until those experiments are conducted. We remain agnostic about the dissociation between brain mechanisms and behavior; we simply highlight WEIRD populations as an issue that requires caution when interpreting and generalizing results from both the brain and behavior. We make this much clearer on page 5.

This same type of issue came up in the Economic Decision-Making section in which the ms describes how interacting with a friend is different than a computer. It was not clear how neuroscience was important here or told us anything different than the behavioral data. The ms references that there was enhanced ventral striatal activation but that is simply the same neural region typically associated with reward, so I wasn't sure how that contributed anything beyond what the enhanced subjective ratings told us).

Again, we thank the reviewer for highlighting this weakness. We have re-written the Economic Decision-Making section such that we now focus on distinctions between activation patterns and behavior in tasks involving other people versus tasks involving computers (see pages 7-9). The re-focusing alleviates the issue described by the reviewers, and gives us an opportunity to highlight the contribution of the brain activity data over and above the behavioral data.

The Economic Decision-Making section (and also the Contribution of Social Neuroscience to Health) seemed focused on stating why social context and emotions needed to be taken into account but it wasn't clear how this was related to the thesis of showing how neuroscience contributes to psychology. Instead it seemed like a case for how psychology contributes to economics or health and no brain data was needed. A quick review of topic sentences in those sections illustrate that the writing doesn't even seem to be focused on anything about neuroscience's contribution.

We have re-written both sections (see pages 6-16) to clearly spell out the contribution of the neuroscience to the psychology.

I found it surprising that the paper recommends that social neuroscientists should avoid using deception when published data has shown that deception doesn't make a difference in behavior.

We highlight the use of deception only in terms of the standards in the economics literature, and a potential concern if researchers wished to engage in conversation with the academic discipline. As such, we are not advocating a deception free psychology, but simply highlighting a problem for translational research. We state this explicitly on pages 9-10.

The paper rightfully raises reverse inference as a problem but then falls victim to it. For example, on p. 4, the ms notes that caudate activation indicates that managers were using a heuristic decision strategy appropos of nothing about the actual manipulation but instead based on viewing caudate as a marker of automaticity. So I didn't understand how these examples help us learn about how neuroscience contributes to our knowledge of decision-making.

We thank the reviewer for highlighting our inconsistency regarding reverse inferences. While we rightfully raise it as a concern, a lot of literature in neuroeconomics still relies on reverse inferences to draw conclusions. This is not perceived as a problem perhaps because there is less debate about the function of particularly brain regions like the striatum that are commonly engaged in economics tasks. This allows us to have a larger conversation about reverse inferences that is more nuanced than simply negative (see page 10, 15, 18, 21-22). Moreover, throughout the manuscript we removed sentences where we use reverse inferencing to draw conclusions about previous fMRI data.

These types of interpretation and focus problems are riddled through the mansucript. My sense is that there is a lot of useful information here. A revision should focus on stating in what ways *social neuroscience*  contributes (or has tried to contribute and failed) to these domains rather than discussing how social and emotional contexts are important.

Again, we thank the reviewers for their insightful comments. We feel like our revised manuscript is now more focused on these contributions and no longer riddled with interpretation and focus problems.

Reviewer 2 Report

The authors present a manuscript in which they aim to “review the contribution of Social Neuroscience” to three fields, Economics, Health and Law.  They state that they will “address the concerns that the extrapolation of neuroscientific results to applied issues raises within each of these domains, and […] suggest guidelines and good practices to circumvent these concerns”. The topic is clearly of great interest to society (and, as a result, to readers), however, I do not feel that the current manuscript adequately discusses the topic or potential solutions. Therefore, I cannot recommend it for publication at this time. Below I detail with more specificity why I feel this is the case.

1)  General Clarity
The manuscript as written needs a lot of work to both correct the grammar (examples too numerous to detail), as well as to ensure that the correct words are being used (e.g. Line 274 I believe “causal influences” should read “causal inferences”; Line 223 I believe “executive control functions” should read “executive control regions”).  On top of this, there are a number of places in which the sentences and/or arguments are not clear (e.g. Lines 297-298, what does mechanizing a person mean?; Lines 407-408, what does the “adequacy of the samples” pertain to?).  The manuscript needs to be carefully reviewed to ensure that the arguments make logical sense and that they contain enough information so as to be understood by the reader (e.g. what is the trolley problem – this reviewer knows, but should a standard reader be expected to?).

2) Reverse Inference
While the authors do raise the problem of reverse inference (e.g. lines 47-67, 168-174, etc.), they then turn around and use reverse inference in a number of their pieces of evidence for the contribution of social neuroscience (e.g. lines 135-138, lines 182-183, lines 262-264, lines 307-314).  Furthermore, one solution they suggest can address issues of reverse inference is flawed (Lines 401-405: existing automated meta-analysis tools such as Brainmap and Neurosynth rely on researcher identification of psychological processes of interest and as such suffer from circular reasoning that even their creators acknowledge), and another (Lines 174-177: using an independent localizer task) does *not* address the problem of reverse inference at all – though it does address the issue of circular reasoning (AKA double dipping).  This raises serious issues with the interpretation of the arguments presented in the review.

3) Issues of sample size and replicability
There are major issues pertaining to replicability, methodological negligence, and statistical power that are currently being addressed in psychology in general (and really science as a whole), and in the field of social psychology/neuroscience in particular. The authors do not address this at all though it would be incredibly relevant to those seeking to understand whether previous social neuroscience findings were reliable enough to be considered applicable to real world problems at this juncture.

4) The authors need to do a better job of demonstrating that social neuroscience findings provide much above and beyond what behavioral findings would provide (once you’ve accounted for issues with reverse inference).  To be clear, that is not to say that I do not believe that this argument can be made – I just believe the authors need to make it more clearly and explicitly.

5) Lines 102-105: The sentence “As neuroeconomists have increasingly incorporated insights from the field of social psychology and social neuroscience they have now shifted away from the simplified idea that people always make rational and self-interested decisions as a result of careful deliberation” is not really true as far as I understand it.  Most economists still believe that people make rational, self-interested decisions, however, the value functions over which they are optimizing are more complex and more subjective than originally thought.

 Because of the issues listed above, I regretfully cannot recommend this manuscript for publication at this time.

Author Response

Reviewer 2

The authors present a manuscript in which they aim to “review the contribution of Social Neuroscience” to three fields, Economics, Health and Law.  They state that they will “address the concerns that the extrapolation of neuroscientific results to applied issues raises within each of these domains, and […] suggest guidelines and good practices to circumvent these concerns”. The topic is clearly of great interest to society (and, as a result, to readers), however, I do not feel that the current manuscript adequately discusses the topic or potential solutions. Therefore, I cannot recommend it for publication at this time. Below I detail with more specificity why I feel this is the case.

We thank the reviewer for their somewhat positive response to our manuscript and hope the suggested changes now make our manuscript more suitable for publication.

1)  General Clarity

The manuscript as written needs a lot of work to both correct the grammar (examples too numerous to detail), as well as to ensure that the correct words are being used (e.g. Line 274 I believe “causal influences” should read “causal inferences”; Line 223 I believe “executive control functions” should read “executive control regions”).  On top of this, there are a number of places in which the sentences and/or arguments are not clear (e.g. Lines 297-298, what does mechanizing a person mean?; Lines 407-408, what does the “adequacy of the samples” pertain to?).  The manuscript needs to be carefully reviewed to ensure that the arguments make logical sense and that they contain enough information so as to be understood by the reader (e.g. what is the trolley problem – this reviewer knows, but should a standard reader be expected to?).

We thank the reviewer for pointing out these grammatical errors. We have since re-written the manuscript, and proofread, so we hope all grammatical errors are corrected. We have also specifically focused on the examples highlighted by the reviewer, and if those sentences have been retain, they have been corrected.

2) Reverse Inference

While the authors do raise the problem of reverse inference (e.g. lines 47-67, 168-174, etc.), they then turn around and use reverse inference in a number of their pieces of evidence for the contribution of social neuroscience (e.g. lines 135-138, lines 182-183, lines 262-264, lines 307-314).  Furthermore, one solution they suggest can address issues of reverse inference is flawed (Lines 401-405: existing automated meta-analysis tools such as Brainmap and Neurosynth rely on researcher identification of psychological processes of interest and as such suffer from circular reasoning that even their creators acknowledge), and another (Lines 174-177: using an independent localizer task) does *not* address the problem of reverse inference at all – though it does address the issue of circular reasoning (AKA double dipping).  This raises serious issues with the interpretation of the arguments presented in the review.

We have addressed the issue of reverse inferences in our response to the reviewer above. Moreover, we have now taken a more cautious approach (see pages 10, 15, 18, 21-22)

3) Issues of sample size and replicability

There are major issues pertaining to replicability, methodological negligence, and statistical power that are currently being addressed in psychology in general (and really science as a whole), and in the field of social psychology/neuroscience in particular. The authors do not address this at all though it would be incredibly relevant to those seeking to understand whether previous social neuroscience findings were reliable enough to be considered applicable to real world problems at this juncture.

We thank the reviewer greatly for identifying this oversight on our behalf. We now address the replication crisis and other statistical and methodological concerns common to all of science on pages 5-6 and 22.

4) The authors need to do a better job of demonstrating that social neuroscience findings provide much above and beyond what behavioral findings would provide (once you’ve accounted for issues with reverse inference).  To be clear, that is not to say that I do not believe that this argument can be made – I just believe the authors need to make it more clearly and explicitly.

We agree with the reviewer and have responded to a similar comment from the other reviewer above. In a nutshell, we believe that the brain can serve as an index of psychological processing, even when considering the challenges provided by reverse inferences (e.g. see page 22). This argument highlights the utility of neuroscience above and beyond behavioral data.

5) Lines 102-105: The sentence “As neuroeconomists have increasingly incorporated insights from the field of social psychology and social neuroscience they have now shifted away from the simplified idea that people always make rational and self-interested decisions as a result of careful deliberation” is not really true as far as I understand it.  Most economists still believe that people make rational, self-interested decisions, however, the value functions over which they are optimizing are more complex and more subjective than originally thought.

We have edited substantially our section on economics. We agree that people in that discipline still believe that human beings are rational, but indeed, as the reviewer suggests, such experts are now more willing to integrate information about actual human behavior that accounts for social factors such as identity and the social context (see page 6).

Round 2

Reviewer 1 Report

Thank you for addressing the points I raised, this article will be a valuable resource for the field

Author Response

We would like to thank the reviewer for his suggestions and comments, which helped us improve the clarity and precision of our manuscript.

Reviewer 2 Report

The authors present a revision of their original article reviewing the contributions of social neuroscience to the fields of economics, health, and law.  My primary concerns with the first draft (many of which were shared with the other reviewer) were 1) overall clarity, 2) the use of reverse inference, 3) the lack of a discussion about issues of sample size and replicability, and 4) a lack of clear evidence that social neuroscience had contributed to each field above and beyond what behavioral studies could. In reviewing this resubmission, I focused on these 4 points.

While I do appreciate that the authors made a serious attempt at revising the manuscript, and it is in much better shape than the first submission, it still does not appear to me to be ready for publication.  Given the current blossoming of these fields, I feel that the authors could do a better job of justifying the power of the social neuroscience approach (even if it’s just promise for the future).  As is, the manuscript does not leave the reader convinced and in a few places still suffers from reverse inference and lack of clarity.  I detail some specific issues below.

1)    Overall Clarity: The overall clarity of the manuscript is quite improved (there are still a few grammatical errors, e.g. “experimental demands effects” (page 2, line 73)), though there are still a few places where the authors’ intent/logic remains unclear - many of which pertain to reverse inference (see below)

2)    Reverse inference:

a.     Page 2, paragraph on reverse inference: The authors define what reverse inference is, but do not clearly state why reverse inference is a problem. The reasons that they provide are inaccurate and/or unclear. In particular, they imply that one reason that reverse inference is “more acute” in applied social domains is that they are “…highly complex mental phenomena that are unlikely to be implemented in one single brain system.” as if implementation in a singular brain system was a prerequisite for problematic reverse inference (it is not). They then offer that complexity and context dependence is another reason why reverse inference is more acute (however they do not explain how).  Problematic reverse inference results from a logical fallacy to which many fMRI studies fell victim (and rigorous attempts at reverse inference avoid this logical fallacy).  The authors need to better explain their logic for why social neuroscience was/is more susceptible to problematic reverse inference.

b.     Page 5, lines 202-205: The authors suggest that neuroeconomics (uniquely?) offers solutions to the problem of reverse inference, one of which is implementing “theory-driven experimental design by directly manipulating the underlying process”. I do not see how this is a solution to problematic reverse inference (which again – is a logical fallacy – brain activity given psychological process does not translate to psychological process given brain activity). Moreover, theoretically-driven experimental design is not unique (nor derived from) neuroeconomics.

c.     The authors still have some unwarranted reverse inference embedded in the manuscript on page 7, lines 304-308: “research show that when participants are encouraged to individuate targets, activation in the […] (MPFC) […] increased even when viewing homeless people and drug addicts …”. That there was increased activity in a “mentalizing” region such as the MPFC cannot be taken as evidence of increased mentalizing.

d.     The first paragraph of section 4.1 (on page 8) contains multiple instances of unwarranted reverse inference (e.g. that activity in the amygdala is indicative of an affective response, that amygdala activity signals a communicative function, that the involvement of systems other than the amygdala is indicative of functions other than communication).

e.     Page 9, line 398: The language is too strong here – “demonstrate” should be replaced with “suggest” and “decisions rely on” should be changed to “decisions may rely on”.

f.      Page 9, line 414-415: It is unclear what the authors mean by “cognitive information” – and it is unclear that inferring an interaction between emotional information and cognitive information from imaging results is not problematic in this case.

3)    Evidence of the unique contribution of social neuroscience:  Overall I still don’t feel that the manuscript makes a very strong argument for a unique contribution to each of the fields of Economics, Health, and Law. In particular, the section on social neuroscience and health (page 6) is not very strong and needs clarification and strengthening of the argument that social neuroscience has made a significant contribution.  It’s unclear that social neuroscience was needed to determine that “the perception of social threat, rather than actual threat, seems to determine people’s sensitization to rejection…” or that “social perceptions by health care professionals may shape their interactions with patients, and accordingly, the treatment patients receive”.  The onus is on the authors to demonstrate that social neuroscience has contributed something above and beyond what a behavioral/survey study could have contributed. I am not a skeptic, I just feel that their argument isn’t strong here and could be strengthened throughout the manuscript.

Other Minor Comments:

1)    Given the prominence of economic games utilized in social paradigms, it would be worthwhile for the authors to provide an example of one such game when they introduce the concept on page 3.

2)    Page 3, lines 137-138: It’s unclear to me how the endowment effect is related to reward processing per se (valuation, yes, but reward processing?).

Author Response

Thank you for the opportunity to revise our manuscript. Below, we respond to reviewer 2’s comments in detail. The reviewer’s comments are italicized, while our responses remain in standard, non-italicized font.

Reviewer 2

1)    Overall Clarity: The overall clarity of the manuscript is quite improved (there are still a few grammatical errors, e.g. “experimental demands effects” (page 2, line 73)), though there are still a few places where the authors’ intent/logic remains unclear - many of which pertain to reverse inference (see below)

We would like to thank the reviewer for having pointed to the few grammatical errors our manuscript entailed. We have now revised it and corrected them. We address the reviewer’s comments on reverse inference in the paragraphs below.

2)    Reverse inference:

a.     Page 2, paragraph on reverse inference: The authors define what reverse inference is, but do not clearly state why reverse inference is a problem. The reasons that they provide are inaccurate and/or unclear. In particular, they imply that one reason that reverse inference is “more acute” in applied social domains is that they are “…highly complex mental phenomena that are unlikely to be implemented in one single brain system.” as if implementation in a singular brain system was a prerequisite for problematic reverse inference (it is not). They then offer that complexity and context dependence is another reason why reverse inference is more acute (however they do not explain how).  Problematic reverse inference results from a logical fallacy to which many fMRI studies fell victim (and rigorous attempts at reverse inference avoid this logical fallacy).  The authors need to better explain their logic for why social neuroscience was/is more susceptible to problematic reverse inference.

We are grateful to the reviewer for having pointed to this lack of clarity. In the new version of our manuscript, we have now considerably revised the section of the introduction on reverse inference (see p. 2, lines 47-82) to address these concerns.

b.     Page 5, lines 202-205: The authors suggest that neuroeconomics (uniquely?) offers solutions to the problem of reverse inference, one of which is implementing “theory-driven experimental design by directly manipulating the underlying process”. I do not see how this is a solution to problematic reverse inference (which again – is a logical fallacy – brain activity given psychological process does not translate to psychological process given brain activity). Moreover, theoretically-driven experimental design is not unique (nor derived from) neuroeconomics.

In the previous version of our article, we did not mean to suggest that neuroeconomics uniquely offers solutions to the problem of reverse inference but we understand that our formulation may have been confusing. Therefore, we have rephrased this sentence on p. 5, line 226.

Concerning the question of theory-driven experimental designs, we agree with the reviewer that it does not offer a solution to the problem of reverse inference; however, we believe that it constitutes an important methodological consideration to take into account in order to uncover specific associations between brain activity and cognitive functions. We develop this argument on p. 5, lines 228-231.

c.     The authors still have some unwarranted reverse inference embedded in the manuscript on page 7, lines 304-308: “research show that when participants are encouraged to individuate targets, activation in the […] (MPFC) […] increased even when viewing homeless people and drug addicts …”. That there was increased activity in a “mentalizing” region such as the MPFC cannot be taken as evidence of increased mentalizing.

We agree with the reviewer and are now more careful with interpreting fMRI results in our article. To that aim, we have revised the paragraphs mentioned by the reviewer as containing unwarranted reverse inferences.

In addition, we chose to replace the paragraph mentioned by the reviewer above with recent evidence that more clearly illustrates how social neuroscience can aid in the development of interventions to reduce biases that may be less easily assessed using self-reports or behavioral measures (see also Reviewer 2's comment 3). We discuss findings of two separate studies, one using fMRI and one using EEG, which demonstrate a racial bias in pain perception (in the absence of clear behavioral evidence) and that examine the effectiveness of different strategies (intergroup contact; recategorization) to reduce these biases (see p. 7, lines 326-345).

d.     The first paragraph of section 4.1 (on page 8) contains multiple instances of unwarranted reverse inference (e.g. that activity in the amygdala is indicative of an affective response, that amygdala activity signals a communicative function, that the involvement of systems other than the amygdala is indicative of functions other than communication).

As mentioned in the previous section, we are now more careful with interpreting fMRI results and we have revised the paragraphs containing unwarranted reverse inferences (see p. 9, lines 407-423).

e.     Page 9, line 398: The language is too strong here – “demonstrate” should be replaced with “suggest” and “decisions rely on” should be changed to “decisions may rely on”.

We would like to thank the reviewer for his suggestions that we have implemented in the revised version of the manuscript (see p. 9, lines 427 and 432).

f.      Page 9, line 414-415: It is unclear what the authors mean by “cognitive information” – and it is unclear that inferring an interaction between emotional information and cognitive information from imaging results is not problematic in this case.

We have revised these sentences and, to improve clarity and we have replaced the expression “cognitive information” by “legal reasoning” (see p. 10, lines 446-454).

3)    Evidence of the unique contribution of social neuroscience:  Overall I still don’t feel that the manuscript makes a very strong argument for a unique contribution to each of the fields of Economics, Health, and Law. In particular, the section on social neuroscience and health (page 6) is not very strong and needs clarification and strengthening of the argument that social neuroscience has made a significant contribution.  It’s unclear that social neuroscience was needed to determine that “the perception of social threat, rather than actual threat, seems to determine people’s sensitization to rejection…” or that “social perceptions by health care professionals may shape their interactions with patients, and accordingly, the treatment patients receive”.  The onus is on the authors to demonstrate that social neuroscience has contributed something above and beyond what a behavioral/survey study could have contributed. I am not a skeptic, I just feel that their argument isn’t strong here and could be strengthened throughout the manuscript.

As described in our response to Comment 2c, in this revised version of the article we emphasize more clearly the contribution of social neuroscience to the applied domains that our article addresses. To that aim we have added a paragraph to the section on social neuroscience of health (see p. 7, lines 326-345) as well as in the discussion (see p. 11, lines 496-513).

Other Minor Comments:

1)    Given the prominence of economic games utilized in social paradigms, it would be worthwhile for the authors to provide an example of one such game when they introduce the concept on page 3.

We are thankful to the reviewer for this suggestion. We now provide a description of the Ultimatum Game in the introduction of the section on neuroeconomics p. 3, lines 136-145.

2)    Page 3, lines 137-138: It’s unclear to me how the endowment effect is related to reward processing per se (valuation, yes, but reward processing?).

We agree with the reviewer and have removed the aforementioned reference to the endowment effect.
